# In Vitro Digestion of Vacuum-Impregnated Yam Bean Snacks: *Pediococcus acidilactici* Viability and Mango Seed Polyphenol Bioaccessibility

**DOI:** 10.3390/microorganisms12101993

**Published:** 2024-09-30

**Authors:** Alba Cecilia Durán-Castañeda, Adela Yolanda Bueno-Durán, Manuel Iván Girón-Pérez, Juan Arturo Ragazzo-Sánchez, Jorge Alberto Sánchez-Burgos, Sonia Guadalupe Sáyago-Ayerdi, Victor Manuel Zamora-Gasga

**Affiliations:** 1Instituto Tecnológico de Tepic, Tecnológico Nacional de México, Av. Tecnológico No 2595, Col. Lagos del Country, Tepic CP 63175, Nayarit, Mexico; alceduranca@ittepic.edu.mx (A.C.D.-C.); jragazzo@tepic.tecnm.mx (J.A.R.-S.); jsanchezb@tepic.tecnm.mx (J.A.S.-B.); ssayago@tepic.tecnm.mx (S.G.S.-A.); 2Laboratorio Nacional para la Investigación en Inocuidad Alimentaria (LANIIA)—Unidad Nayarit, Universidad Autónoma de Nayarit, Calle Tres S/N, Colonia Cd. Industrial, Tepic CP 63173, Nayarit, Mexico; abueno.laniia@gmail.com (A.Y.B.-D.); ivangiron@uan.edu.mx (M.I.G.-P.)

**Keywords:** probiotic, mango by-products, antioxidants, gastrointestinal digestion, functional foods, emerging technologies

## Abstract

This study investigates the in vitro digestion of vacuum-impregnated *yam bean* snacks enriched with *Pediococcus acidilactici* and mango seed polyphenols, focusing on bacterial survival and polyphenol bioaccessibility. The snacks were prepared by vacuum impregnation (VI) with solutions containing either mango seed extract, *P. acidilactici*, or a combination of both, followed by dehydration. The antimicrobial activity of the treatments was assessed against pathogens, revealing limited effectiveness, likely due to insufficient concentrations of mango seed extract and the intrinsic resistance of the bacteria. VI of mango seed extract improved the total soluble phenols (TSP) content up to 400% and maintained the initial probiotic concentration (10^6^ cell/mL). In vitro digestion was performed to simulate gastrointestinal conditions, measuring the stability of TSP and the survival of *P. acidilactici.* The results indicated that the viability of *P. acidilactici* fluctuated throughout the digestion process (10^6^ to 10^4^ log UFC/g), the polyphenols showed varying degrees of bioaccessibility (11 to 30%), and the TSP content in the intestinal fraction ranged from 1.95 to 6.54 mg GAE/g. The study highlights the potential of VI for incorporating functional components into plant-based snacks, though further optimization is necessary to enhance the stability of *P. acidilactici* and the effectiveness of the bioactive ingredients.

## 1. Introduction

The increasing demand for foods that not only nourish but also promote health has spurred the development of a new generation of foods enriched with functional ingredients like probiotics and prebiotics [1]. This shift in consumer preferences reflects a growing awareness of the critical role of diet in disease prevention and quality of life improvement. Today’s consumers seek products that go beyond basic nutritional needs, opting for foods that provide additional benefits, such as strengthening the immune system, improving digestive health, and preventing chronic diseases [2].

Functional ingredients like probiotics and prebiotics are highly valued for their ability to positively influence gut microbiota, a crucial aspect of human health that has garnered significant attention in recent years [3]. Probiotics are defined as live microorganisms that, when administered in adequate amounts, confer health benefits to the host. These beneficial bacteria are commonly found in the gastrointestinal tract, where they play a key role in maintaining gut microbiota balance [4,5]. Probiotics are distinguished by their ability to survive the acidic environment of the stomach and the alkaline conditions of the intestine, enabling them to effectively colonize the gut [6]. One notable probiotic strain is *Pediococcus acidilactici*, a lactic acid bacterium widely recognized for its robust ability to thrive in a variety of environmental conditions. *P. acidilactici* is particularly valued for its antimicrobial properties, which are largely attributed to its production of bacteriocins—antimicrobial peptides that inhibit the growth of pathogenic bacteria [7]. Beyond its antimicrobial activity, *P. acidilactici* enhances intestinal barrier function, promotes the production of short-chain fatty acids, and contributes to immune system modulation [8]. These bacteria were previously investigated in a study on the in vitro colonic fermentation dynamics, where non-digestible carbohydrates from yam beans were used as the substrate. The study assessed metabolite generation at various fermentation times and found that the inclusion of *P. acidilactici* in the fermentation system significantly altered the metabolite profile; these results suggest that it can also actively influence the fermentation environment [9].

Prebiotics, which include a wide range of indigestible compounds such as dietary fiber and polyphenols, play a vital role as substrates for probiotic bacteria, supporting their growth and activity in the gut [10]. Polyphenols, in particular, have attracted significant interest due to their dual role as both antioxidants and prebiotics. These naturally occurring compounds are found in a variety of plant-based foods and contribute not only to the nutritional value of these foods but also to the maintenance of a healthy gut microbiota [10,11]. A promising source of polyphenols is the by-products of fruit processing, such as mango seeds. Often discarded as waste, these by-products are rich in polyphenols, with gallotannins being the most abundant group, the most prominent being penta-galloyl glucose (2407.94 mg/100 g). Other significant compounds include gallic acid (1449.27 mg/100 g) and 6-O-galloyl glucose (468.40 mg/100 g). Additionally, xanthones such as mangiferin (36.88 mg/100 g) and flavonoids like quercetin xyloside (2.95 mg/100 g) are present. In total, this by-product contains 6568.61 mg/100 g of phenolic compounds, reflecting their potential as a rich source of bioactive compounds that significantly contribute to health through intestinal biotransformation [12,13,14]. Moreover, the valorization of by-products not only reduces waste but also provides an innovative way to deliver these beneficial compounds in functional foods [15].

Traditionally, probiotics have been incorporated into animal-based foods such as dairy-fermented products [16]. However, there is a growing interest in expanding the inclusion of probiotics in plant-based matrices, not only to cater to vegan consumers or those with food intolerances but also to diversify the sources of these beneficial microorganisms [17]. Notably, legumes have shown potential as substrates for probiotic incorporation. For example, *Lactobacillus acidophilus* and *Lactobacillus casei* were successfully used in the fermentation of pigeon pea (*Cajanus cajan*), producing legume-based fermented products with high viability of probiotic strains throughout storage; these formulations demonstrated good sensory attributes and could serve as a cost-effective source of protein while supporting the growth of probiotics, making them suitable carriers for functional food development [18]. Other studies have also highlighted the successful incorporation of various probiotic strains into legume-based products, further supporting the potential of legumes as effective vehicles for delivering health-promoting microorganisms [19]. In this context, vacuum impregnation (VI) emerges as a promising technology for incorporating functional ingredients into plant-based foods. During the VI process, the food matrix is exposed to vacuum pressure, which expels the air trapped within the pores of the intracellular material, temporarily enlarging these spaces. Upon the restoration of normal pressure, a pressure gradient is created, facilitating the infiltration of these vacated spaces by a liquid known as the impregnating solution. This solution carries the functional components of interest, enabling their uniform incorporation into the food matrix [20,21,22].

The successful application of VI in various plant matrices highlights its potential as a method for incorporating functional ingredients without compromising the sensory or nutritional properties of food [23]. For instance, De Oliveira et al. [24] compared the use of vacuum impregnation and soaking techniques to incorporate the probiotic *Lactobacillus acidophilus* into minimally processed melon. Their findings showed that VI was more effective than soaking in maintaining probiotic viability, with counts comparable to those in probiotic dairy products. Moreover, Burca-Busaga et al. [25] demonstrated the use of VI in apples to enhance the incorporation of bioactive compounds, including probiotics and vitamins, without significantly altering the fruit’s texture or flavor. Additionally, Barrera et al. [26] explored the use of VI in clementine juice inoculated with *Lactobacillus salivarius*, showing that the addition of trehalose and sublethal homogenization significantly enhanced both antioxidant properties and the probiotic effect. One particularly promising matrix for VI is the yam bean (*Pachyrhizus erosus* L.), a tuber native to Mexico belonging to the *Fabaceae* family. This tuber is known for its starchy flavor and porous structure, which makes it well-suited to this process [27]. Due to its natural deficiency in bioactive compounds, the yam bean is an ideal candidate for enrichment through VI with functional components like probiotics and polyphenols. This enrichment not only promotes the valorization of yam beans but also provides new and innovative consumption alternatives [28].

However, a crucial aspect to consider is the stability of these functional components once they have been impregnated [23]. The stability of probiotics, for instance, is essential to ensure that they maintain their viability and effectiveness throughout the product’s shelf life and, more importantly, during their transit through the gastrointestinal tract [29]. Moreover, it is vital to assess the bioaccessibility of the impregnated polyphenols, defined as the proportion of these compounds released and available for absorption during digestion [30]. During the digestive process, foods are subjected to a series of enzymatic reactions and pH changes that can either release or degrade bioactive components, thereby affecting their bioavailability [31]. Consequently, in vitro gastrointestinal digestion studies are valuable tools for predicting how probiotics and polyphenols will behave in the body, enabling a better understanding of their potential health benefits [32]. Therefore, the objective of this study was to evaluate the antimicrobial activity of *P. acidilactici* and mango seed extract bacteria prior to the formulation of the snacks. Subsequently, the study aimed to assess the survival of *P. acidilactici* and the bioaccessibility of mango seed extract in vacuum-impregnated yam bean snacks, followed by an in vitro digestion assessment.

## 2. Materials and Methods

### 2.1. Mango Seed Polyphenol Extraction, Preparation of the Suspension of P. acidilactici and Yam Bean Conditioning

Mango seeds (*Mangifera indica* L. ‘Ataulfo’) were obtained after a depulping process and were placed in a dehydrator oven (Cabelas TS160 D, San Jose, CA, USA) to be dried at 45 °C for 24 h. Subsequently, they were ground and sieved to obtain flour with a particle size of approximately ≤300 μm. Polyphenols were extracted from mango seeds following the procedure outlined by Martínez-Olivo et al. [14]. Briefly, the flour was combined with a water–ethanol solution (80:20) in a 1:35 weight-to-volume ratio (28.57 g of flour with 1000 mL of solution). This mixture was subjected to sonication using a Hielscher UP400S (Teltow, Germany) ultrasonic processor while maintaining the temperature at 60 °C ± 2 °C with a thermal bath (TERLAB TE-840D, Guadalajara, Mexico) for 8 min, maintaining a constant frequency of 24 kHz and an amplitude of 260 μm (40%). The sonotrode used (H40, Hielscher, Teltow, Germany) had a radiating diameter of 40 mm and a power density of 12 W/cm^2^. The resulting extract was filtered and concentrated using a rotary evaporator (BÜCHI Labortechnik, R-100, B-100-F105, Flawil, Germany) at 160 mBar and 60 °C for one hour, with the condenser set to 6 °C. The final concentrate was frozen at −80 °C and lyophilized (FreeZone 6, Labconco, Kansas City, MO, USA) at −45 °C and 0.113 mBar for 48 h. It was then ground and sieved to obtain a particle size of ≤500 μm. The polyphenol powder was stored in vacuum-sealed bags at −20 °C until further use.

*Pediococcus acidilactici* was isolated from a colonic fermentation residue containing a consortium of human fecal inoculum, from which *Enterobacteriaceae*, *Lactobacillus*, *Bifidobacterium*, and *Bacillus* were identified. After purifying the strain, it was sent to the Centro de Investigación y Asistencia en Tecnología y Diseño del Estado de Jalisco (CIATEJ), where taxonomic identification was performed using MALDI-TOF mass spectrometry and the obtained spectrum was compared against the Biotyper^®^ Reference Libraries (BDAL, v.10). The match yielded a score of 2.298, within the range of 2.000 to 2.999, indicating secure genus-level identification.

The activation of the strain was carried out following the protocol by Pi et al. [33], with modifications by Durán-Castañeda et al. [9]. The strain was activated in MRS broth and incubated for 7 h at 37 °C under anaerobic conditions. The viable cell count was determined using a Neubauer chamber with trypan blue staining, and the inoculum was adjusted to 10^6^ cells/mL. The culture was centrifuged at 6000 rpm for 10 min at 25 °C (Hermle Z 323 K; Wehingen, Germany). The pellet was washed twice with peptone water and then resuspended in the same solution. The suspension was refrigerated until further use.

Milky yam beans (*Pachyrhizus erosus* L.) were purchased at consumption maturity from a local market in Tepic, Nayarit, Mexico (21.5095°, −104.89569°), during the Spring–Summer harvest cycle of 2023, with an average weight of 500 ± 100 g and uniform shape. The tubers were washed by immersion in purified water, and organic residues were removed using a nylon-bristle brush. After washing, the tubers were dried at room temperature, and the peel was manually removed. The tubers were sliced using a slicer (Torrey, R-300-A, Guadalajara, Mexico) to obtain slices with a thickness of 1.5 mm and a diameter of 4 cm. These slices were then separated into four different batches (T1, T2, T3, and T4).

### 2.2. Evaluation of the In Vitro Antimicrobial Activity of Mango Seed Extract and P. acidilactici Suspension against Pathogenic Bacteria

In vitro, antimicrobial activity was evaluated using the Kirby–Bauer disk diffusion method. Pathogenic strains (*Staphylococcus aureus*, *Escherichia coli*, *Pseudomonas aeruginosa*, and *Serratia marcescens*) were activated in Luria broth and incubated at 35 ± 2 °C for 14 ± 2 h. Mueller–Hinton agar plates were prepared, and a suspension of each pathogenic bacterium, adjusted to 0.5 McFarland standard [34], was inoculated. Blank disks (6 mm diameter) were loaded in triplicate with 20 µL of mango seed extract reconstituted in sterile distilled water at a concentration of 7 mg/mL. Another triplicate was loaded with 20 µL of *P. acidilactici* suspension at 10^6^ cell/mL, and a final triplicate with 20 µL of a 1:1 (*v*/*v*) combination of mango seed extract (7 mg/mL) and *P. acidilactici* (10^6^ cell/mL), resulting in a final concentration of 3.5 mg/mL for mango seed extract and approximately 10^5^–10^6^ cell/mL for *P. acidilactici*. The three treatments were tested against the four pathogenic bacteria by placing the disks on Mueller–Hinton agar plates with inoculated bacteria. Additionally, antibiotic disks were used as positive controls: tetracycline (30 mg for *E. coli*), gentamicin (10 mg for *P. aeruginosa* and S. aureus), and cefotaxime (30 mg for *S. marcescens*). The plates were incubated at 35 ± 2 °C for 24 ± 2 h, and inhibition zones were measured with calipers. The results were compared to Clinical and Laboratory Standards Institute (CLSI) guidelines [35] to classify pathogens as resistant (R), intermediate (I), or susceptible (S). The inhibition zones were classified as follows: gentamicin (10 mg), ≥15 mm (S), 13–14 mm (I), ≤12 mm (R); tetracycline (30 mg), ≥15 mm (S), 12–14 mm (I), ≤11 mm (R); cefotaxime (30 mg), ≥23 mm (S), 20–22 mm (I), ≤19 mm (R). The pathogenic strains and antimicrobial disks were provided by the Laboratorio Nacional para la Investigación en Inocuidad Alimentaria, Unidad Nayarit.

### 2.3. Preparation of Impregnating Solutions and Vacuum Impregnation (VI) Process

Four impregnating solutions were prepared prior to the VI process. T1 served as the control and consisted solely of sterile distilled water. T2 was prepared with mango seed extract reconstituted in sterile distilled water at a concentration of 7 mg/mL. In T3, a suspension of *P. acidilactici* in peptone water was used at a concentration of 10^6^ cell/mL. T4 consisted of a 1:1 (*v*/*v*) combination of mango seed extract (7 mg/mL) and *P. acidilactici* suspension (10^6^ cell/mL), resulting in final concentrations of 3.5 mg/mL for mango seed extract and approximately 10^5^–10^6^ cell/mL for *P. acidilactici*. The quantity of impregnating solution prepared was determined based on a 3:1 (*v*/*w*) ratio of impregnating solution to fresh yam bean slices. All solutions were prepared immediately before use to ensure component stability and activity.

The VI process was performed as described by González-Moya et al. [36] using a vacuum chamber (Bacoeng P0666, Richfield, MT, USA). Vacuum pressure was applied using a vacuum pump (Bacoeng P0652, Richfield, MT, USA) at 66 mBar, with a vacuum time of 15 min and a restoration time of 5 min. After vacuum impregnation, all four treatments were dehydrated (Cabelas TS160 D, San Jose, CA, USA) at 40 ± 0.5 °C for 10 h. After dehydration, the treatments were stored in food-grade bags and refrigerated at 4 °C for further analysis.

### 2.4. Characterization of Vacuum-Impregnated Yam Bean Snacks

#### 2.4.1. Quantification of Total Soluble Phenols (TSP) and Hydrolyzable Polyphenols (HP)

The extraction of phenolic compounds was performed following the methodology described by Pérez-Jiménez et al. [37]. Treatments (250 mg) obtained from Section 2.3 were mixed with 10 mL of a water–methanol solution (50:50, *v*/*v*) acidified with 0.8% hydrochloric acid. This mixture was subjected to orbital shaking (Orbital Shaker, model Reax2, Heidolph Instruments, Schwabach, Germany) for 1 h at 16 rpm. The resulting extracts were centrifuged at 6000 rpm for 10 min at 4 °C (Hermle, Z32HK, Labortechnik GmbH, Wehingen, Germany). The supernatants were collected, and the remaining residues were re-extracted with 10 mL of an acetone solution (70:30, *v*/*v*) for 60 min and centrifuged at the same conditions. Finally, the supernatants were combined in a 25 mL flask containing an acidified methanol–acetone solution (50:50, *v*/*v*). Each extract was reacted with 7.5% (*w*/*v*) sodium carbonate and Folin–Ciocalteu reagent, following the modified protocol by Alvarez-Parrilla et al. [38] for TSP quantification. The absorbance was measured at 750 nm using a microplate reader (Biotek^®^ Synergy HT, Santa Clara, CA, USA) equipped with Gen5 software. Gallic acid was used as the standard, and the results were expressed as milligrams of gallic acid equivalents (GAE) per gram of dry substrate weight (DW).

The HP content was determined using the method described by Hartzfeld et al. [39]. The residues obtained from the aqueous extraction were dispersed in 10 mL of methanol and 1 mL of sulfuric acid (H₂SO₄, 36%, *v*/*v*). The mixture was incubated in a shaking water bath at 70 °C for 22 h, followed by centrifugation at 6000 rpm at 4 °C. The supernatants were collected, and HP quantification was performed using the same procedure as for TSP quantification.

#### 2.4.2. Antioxidant Capacity (AOX)

Antioxidant capacity was evaluated with three methods. The ABTS assay was conducted as described by Re et al. [40]. A solution of ABTS (7 mM) was prepared by dissolving it in potassium persulfate (2.42 mM) and maintaining it in the dark for 14 h. The ABTS solution was then diluted with phosphate buffer until it reached an absorbance of 0.80 ± 0.02, measured at 734 nm using a microplate reader. A 20 µL aliquot of each aqueous-organic extract was mixed with 255 µL of the diluted ABTS solution. The AOX was determined by interpolating the absorbance values against a Trolox standard curve (37.5–600 μM/mL, R^2^ = 0.9997). The results were expressed as μmol Trolox equivalents per gram (μmol TE/g). The radical scavenging activity was assessed by the reduction in the DPPH radical (2,2-diphenyl-1-picrylhydrazyl) at 517 nm using a microplate reader, following the method of Prior et al. [41] with modifications by Alvarez-Parrilla et al. [38]. A 20 µL aliquot of the aqueous-organic extracts was mixed with 200 µL of a 190 µM methanolic DPPH solution for 10 min. The results were expressed as μmol Trolox equivalents per gram (μmol TE/g) using a calibration curve (37.5–600 μM/mL, R^2^ = 0.9995). The ferric-reducing antioxidant power (FRAP) assay was performed according to the method of Benzie and Strain (1996), with modifications by Alvarez-Parrilla et al. [42]. The FRAP solution was prepared by mixing sodium acetate buffer (0.3 M, pH 3.6), TPTZ-HCl (10 mM in 40 mM HCl), and ferric chloride hexahydrate (20 mM) in a 10:1:1 (*v*/*v*/*v*) ratio, and then heated to 37 °C before use. An aliquot of 24 µL of the extracts was mixed with 180 µL of the FRAP solution, and the absorbance was measured at 595 nm after 30 min using a microplate reader (Biotek, Synergy HT). The results were expressed as μmol Trolox equivalents per mL gram (μmol TE/g) using a calibration curve (0.0081–0.13 mM/mL, R^2^ = 0.9997).

#### 2.4.3. Viability of *P*. *acidilactici* after Vacuum Impregnation Process and Dehydration

The count of *P. acidilactici* was performed following the methodology proposed by Akman et al. [43] in vacuum-impregnated samples before and after the dehydration process. One gram of sample was mixed with 9 mL of sterile peptone water and homogenized (Nutribullet, NBR-0804B, Los Angeles, CA, USA) for 1 min. Serial dilutions were prepared up to 10^−6^ logarithmic inversions with sterile water, and the samples were inoculated on selective MRS agar at 37 °C for 48 ± 2 h. After incubation, colony counting was performed, and the results were reported as a logarithm of colony-forming units per gram of food (Log CFU/g) with a detection limit of 10 CFU/mL of sample.

### 2.5. In Vitro Gastrointestinal Digestion of Vacuum-Impregnated Yam Bean Snacks

The dehydrated treatments were subjected to an in vitro gastrointestinal digestion model, adapted from the standardized methodology by the INFOGEST international network [44], coupled with a dialysis process to estimate the bioaccessibility percentage of phenolic compounds (Figure 1). In the oral phase, 1 g of the sample was incubated under constant agitation with salivary amylase (10 mg/mL, A1031, Sigma-Aldrich, Burlington, MA, USA) and simulated salivary fluid (1 M, pH 7, 37 °C, 2 min). The product from this stage was then kept under constant agitation during the gastric phase by adding pepsin (30 mg/mL, P7000, Sigma-Aldrich, Burlington, MA, USA) and lipase (100 mg/mL, L-3126, Sigma-Aldrich, Burlington, MA, USA) in simulated gastric fluid, adjusting the pH to 3 (1 M, pH 3, 37 °C) for 2 h. To digest proteins, lipids, and starch, the gastric chyme was adjusted to a pH of 7 and subjected to third enzymatic hydrolysis under constant agitation by adding pancreatin (134 mg/mL, P-1750, Sigma-Aldrich, Burlington, MA, USA) and bile salts (200 mg/mL, B-8631, Sigma-Aldrich, Burlington, MA, USA) in simulated intestinal fluid (1 M, pH 7, 37 °C) for 2 h to simulate the intestinal phase. After this period, the hydrolyzed sample was designated as the intestinal fraction (IF). The IF was centrifuged (30 min, 4 °C, 4000 rpm) to obtain two fractions: the soluble indigestible fraction (SIF), corresponding to the supernatant, and the insoluble indigestible fraction (IIF), corresponding to the precipitate. The IIF was washed twice with distilled water (5 mL), and the supernatants were combined with the SIF to a determined volume (50 mL). The SIF was subjected to a dialysis process in an aqueous medium using a dialysis membrane (6 h, 12–14 KDa, Sigma-Aldrich, D9652).

All fractions obtained (oral phase, gastric phase, IF, SIF, and IIF) were characterized using the methodologies described in Section 2.4 to evaluate the total soluble phenols (TSP) and hydrolyzable polyphenols (HP) associated with each fraction, their antioxidant capacity (AOX), and the viability of *P. acidilactici*. Once TSP values were obtained, the bioaccessibility of polyphenols (%BA) (1), non-bioavailable polyphenols (NBP, mg GAE/g DW) (2), and potentially bioavailable polyphenols for absorption (PBPA, mg GAE/g DW) (3) were calculated. All results were based on in vitro parameters, using equations proposed by Cruz-Trinidad et al. [45]:(1)$$\%BA=\frac{{TSP}_{IF}}{{TSP}_{IF}+{HP}_{IF}}x\times 100,$$
where %BA is the percentage of bioaccessibility; *TSP_IF_* is the total soluble phenols released in the intestinal fraction; *PH_IF_* is the hydrolyzable polyphenols of the intestinal fraction.
(2)$$NBP={TSP}_{SIF}+{TSP}_{IIF}+{HP}_{IIF},$$
where NBP is the non-bioavailable polyphenols; *TSP_SIF_* is the total soluble phenols released in the soluble indigestible fraction; *TSP_IIF_* is the total soluble phenols released in the insoluble indigestible fraction; *PH_IIF_* is the hydrolyzable polyphenols of the insoluble indigestible fraction.
(3)$$PBPA={TSP}_{IF}+{HP}_{IF}-NBP$$
where PBPA is the potentially bioavailable polyphenols for absorption; *TSP_IF_* is the total soluble phenols released in the intestinal fraction; *HP_IF_* is the total soluble phenols released in the intestinal fraction; *NBP* is the non-bioavailable polyphenols.

### 2.6. Statistical Analysis

All analyses were performed in triplicate (*n* = 3), and the results were expressed as means ± standard deviation (excluding Section 3.1). Data were subjected to one-way ANOVA, and significant differences between means were evaluated using the Fisher LSD test with a significance level of α = 0.05. Data were processed in Statistica software version 10 (Stat Soft. Inc., Tulsa, OK, USA).

## 3. Results and Discussion

### 3.1. Antimicrobial Activity of Mango Seed Extract and P. acidilactici Suspension

Table 1 presents the results associated with the antimicrobial activity of mango seed extract, *P. acidilactici* suspension, and their combination against pathogenic bacteria. The mango seed extract exhibited limited antimicrobial activity against all the tested bacteria. In *S. aureus*, *E. coli*, and *S. marcescens*, the extract produced inhibition zones of 9 mm, which were classified as resistant. In *P. aeruginosa*, a slightly higher sensitivity was observed with an inhibition zone of 10 mm, although it still fell within the resistant category. The combination of mango seed extract with *P. acidilactici* did not significantly enhance antimicrobial activity, resulting in inhibition zones of 6 mm to 7 mm for the evaluated bacteria, which were also classified as resistant. In contrast, the *P. acidilactici* suspension alone showed no antimicrobial activity, with inhibition zones of 0 mm for all the bacteria tested, indicating total resistance to this treatment.

The antibiotics showed significantly higher antimicrobial activity; gentamicin was effective against *S. aureus* and *P. aeruginosa*, 15 mm and 19 mm, respectively). Tetracycline inhibited *E. coli* with a 20 mm zone, and both were classified as susceptible. Cefotaxime, tested against *S. marcescens*, had a 20 mm zone, classified as intermediate resistance.

The resistance response of pathogens to mango seed extract may be associated with the concentration of bioactive compounds present in the mango seed extract that might have been insufficient to exert significant antimicrobial action. Although a concentration of 7 mg/mL was used, it is possible that some of the active compounds in the extract are present in amounts too low to effectively inhibit bacterial growth [46]. Additionally, the nature of the bioactive compounds in mango seed, which may include gallotannins, flavonoids, xanthones, and other antioxidants, might be more geared towards antioxidant activities than antimicrobial ones, thus limiting their effectiveness against pathogenic bacteria [47]. Another possible cause could be the intrinsic resistance of the tested bacteria. Pathogens such as *P. aeruginosa* and *S. marcescens* are known for their ability to resist many antimicrobial agents due to factors like the presence of efflux pumps, the low permeability of their outer membrane, and the production of inactivating enzymes [48,49]. The lack of effectiveness of the mango seed extract might be related to the inability of its bioactive compounds to penetrate these barriers or to withstand the action of bacterial enzymes.

In the case of *P. acidilactici*, the absence of antimicrobial activity might be attributed to its nature as a lactic acid bacterium. Although some lactic acid bacteria can produce antimicrobial compounds such as bacteriocins, it is possible that *P. acidilactici* does not produce sufficient quantities of these compounds under the experimental conditions used or that the produced compounds are not effective against the selected pathogens [8]. Moreover, competition between *P. acidilactici* and the pathogenic bacteria for available nutrients in the medium could have further limited the treatment’s ability to inhibit pathogen growth [50]. Additionally, the lack of observed antimicrobial activity could be due to the method employed. The disk diffusion method used in this study may not be the most suitable for assessing the antimicrobial activity of lactic acid bacteria against pathogens. It is generally recommended that broth assays or agar spot methods be employed for this purpose, as these techniques can better capture the production and diffusion of antimicrobial compounds by lactic acid bacteria [51]. The limitations of the disk diffusion method might have led to an underestimation of the potential antimicrobial effects of *P. acidilactici*, emphasizing the need for alternative testing approaches to accurately assess its inhibitory capabilities.

The combination of mango seed extract with *P. acidilactici* did not significantly enhance antimicrobial activity, which could be due to the lack of synergy between the bioactive compounds of the extract and the mechanisms of action of *P. acidilactici*. It is possible that the compounds in the mango seed extract are not complementary to the potential antimicrobial metabolites produced by *P. acidilactici*, resulting in antimicrobial activity that does not surpass that of the individual components [52]. Furthermore, the reduced effectiveness of the combination compared to the mango seed extract alone could be due to the dilution effect when mixed in a 1:1 ratio, leading to a lower concentration of the active compounds from the extract, thereby diminishing its overall antimicrobial potential.

### 3.2. Total Soluble Phenol (TSP) Content, Hydrolyzable Polyphenols (HP) and Antioxidant Capacity (AOX) of Vacuum-Impregnated Yam Bean Snacks

The characterization of the yam bean-based snacks after the vacuum impregnation and dehydration process is presented in Table 2. The TSP content indicated significant differences (*p* < 0.05) in treatments T2 and T4 compared to the others (T1 and T3). These treatments exhibited higher TSP content, with T2 having the highest concentration (18.21 mg GAE/g). The primary characteristic of this treatment was the presence of mango seed extract, which was present at a concentration of 7 mg per mL of impregnating solution. In the case of T4, its TSP content was lower than T2 (11.66 mg GAE/g DW), which was attributed to the fact that this treatment contained 50% less mango seed extract concentration, as it shared proportion with *P. acidilactici*. However, its TSP value was still higher than the other treatments. T1 and T3 did not show significant differences; the primary characteristic was the addition of *P. acidilactici* in the impregnating solution (in the case of T3), while T1, being a control treatment, exhibited a value of 3.25 mg GAE/g DW, which represented only the vegetable matrix that was impregnated (yam bean), as the impregnating solution contained only water. These results indicated that the presence of mango seed extract increased TSP content by 400 to 600%. The addition of polyphenolic extracts through vacuum impregnation in matrices with low concentrations has shown modifications in TSP content in other studies. Nawirska-Olszańska et al. [53] reported a 51.4% increase in total phenolic content in blackberry treatments impregnated with polyphenolic extract from apple and pear; the increase, although above 50%, was considered lower because the naturally impregnated vegetable matrix had a high content of phenolic compounds. However, Abalos et al. [20], reported an increase of more than 470% in TSP content in sweet potato samples impregnated with a commercial polyphenolic extract. This suggests that the final TSP content impregnated in the matrix will also depend on the natural content of the food, in which case the yam bean is deficient.

Although T4 had higher TSP values compared to T1 and T3, a decrease in this parameter was observed when compared to T2; this behavior may be associated with possible degradation of polyphenols that interact with the impregnated bacteria, as they may be used as a substrate for their metabolism. However, it is difficult to assert this, as the degradation and utilization process may occur over longer exposure and interaction times [54]. Another possible explanation could be the dilution of the initial concentration of phenolic compounds impregnated in the yam bean slices.

The HP content proportions followed the same pattern as the TSP, with treatments T2 and T3 being the most concentrated (6.35 and 6.41 mg GAE/g DW, respectively) and showing no significant differences (*p* > 0.05). The data obtained for HP content were in similar proportions to those reported by González-Moya et al. [36] in yam bean impregnated with a mango seed polyphenolic extract. The authors reported a lower HP content in the control samples (4.11 mg GAE/g DW) compared to their optimal treatment (12.66 mg GAE/g DW). The increase in TSP and HP content observed in T2 and T4 treatments could be attributed to the hydrodynamic mechanism facilitated by pressure gradients during the vacuum impregnation process. This mechanism involves the exchange of gases or internal fluids previously trapped within the food matrix with the external impregnating solution through the open pores created by the pressure changes [55]. Additionally, the increase in HP levels may also result from the formation of interactions between the polyphenolic compounds and indigestible components of the food matrix, which are then released during acid hydrolysis. These interactions might be unique to the vacuum impregnation process and could contribute to the observed increase in HP content [36,56].

Overall, all treatments showed significant differences (*p* < 0.05) in the three-antioxidant capacity (AOX) assays; however, a correlation pattern with TSP and HP content was observed. The ABTS assay showed the highest activity compared to the other assays, with 99.38, followed by FRAP with 74.62, and 62.37 μmol TE/g DW in DPPH; all of these values are reported in T2. González-Moya et al. [36] also reported higher AOX by the ABTS method in vacuum-impregnated treatments with polyphenolic extracts, followed by FRAP and DPPH. This trend may be related to the affinity of the phenolic compounds in the mango seed extract and the free radical inhibition mechanisms established by the assays performed [57]. The main compounds that make up the mango seed extract belong to the gallate family (gallic acid, pentagalloyl-glucose, 6-O-galloyl-glucose), whose functional groups exhibit electron and proton transfer mechanisms; the primary functional groups are hydroxyl (-OH) [14].

### 3.3. Survival of P. acidilactici after Vacuum Impregnation and Dehydration

Table 3 presents the survival values of *Pediococcus acidilactici* after the processing of yam bean-based snacks. Firstly, it is important to note that the impregnating solutions in treatments T1 and T2 did not contain the suspended bacteria, whereas T3 and T4 had an initial concentration of 10^6^ cell/mL in the impregnating solution at 100% and 50%, respectively. The results observed from the vacuum impregnation effect indicated significant differences between T3 and T4 (*p* < 0.05); however, the number of logarithms of CFU/g in the food remained at 10^6^, which indicated that the impregnation process did not affect the initial concentration of the bacteria. These results are similar to those reported by De-Oliveira et al. [24] in minimally processed melons through VI incorporating *Lactobacillus acidophilus*; the authors reported that the melons remained stable in CFU/g content after VI, as they did not reduce their content over an 8-day period. It has been reported that VI can preserve viability because it can protect probiotic microorganisms from oxygen exposure, in addition to ensuring a uniform distribution of the probiotic within the food matrix [25,58]. However, the effectiveness of vacuum impregnation may depend on factors such as the type of food matrix, the probiotic strain used, and specific processing conditions.

When evaluating the effects of dehydration after VI, a slight increase in bacterial concentration was observed in T3 (3.7 × 10^6^ CFU/g DW); however, the number of logarithms remained at 10^6^. In the case of T4, the concentration remained stable, very similar to that obtained during the VI process. The dehydration process combines heat and mass transfer to reduce the water activity of the food; this process can provide greater microbial and enzymatic stability over time [59]. It is important to mention that the probiotic concentration was preserved, indicating that the process of generating the yam bean-based food through VI allowed the survival of *P. acidilactici*.

### 3.4. Data Associated with In Vitro Gastrointestinal Digestion of Vacuum-Impregnated Yam Bean Snacks and Bioaccessibility

#### 3.4.1. Changes in Total Soluble Phenol (TSP) Content and Bioaccessibility

Figure 2 illustrates the changes in TSP content during the in vitro gastrointestinal digestion process. Compared to the undigested sample, a decrease in TSP content was observed at the beginning of the oral stage across all treatments. T2 and T4 showed the most significant decline; however, they maintained a higher TSP content compared to the other treatments, which is related to the nature of the food matrix and the addition of mango seed extract. During the oral phase, the size of the food particles is reduced, and the hydrolysis of certain complex carbohydrates, such as starch, occurs through the hydrolysis of glycosidic bonds by the action of salivary alpha-amylase [60].

In the gastric phase, TSP content remained stable in T1 and T3, with T1 showing a slight increase, measuring 3.69 mg GAE/g DW and 2.21 mg GAE/g DW, respectively. Conversely, T2 and T4 exhibited less stability, with TSP content decreasing more significantly than in the first stage, resulting in 3.68 mg GAE/g DW for T2 and 2.74 mg GAE/g DW for T4. The instability in the gastric stage is due to the acidic and enzymatic conditions within the stomach environment. The pH can influence the dissociation and degradation of the compounds, and the enzymatic action of pepsin can break down the main bonds of the phenolic structure, reducing the bioaccessibility of the compound [61].

During the intestinal stage, only T2 showed a release of TSP, while the other treatments continued to decrease; these released compounds are associated with the intestinal fraction (IF). A low release of phenolic compounds in the intestinal environment is related to their interaction with indigestible components. These interactions can involve chemical bonds formed natively in the food or through modifications during processing. It has been reported that complex carbohydrates can form glycosidic bonds with phenolic compounds, as well as hydrogen bonds with hydroxyl groups [15]. These interactions lead to a “carryover” effect that reduces the bioaccessibility and bioavailability of phenolic compounds in the small intestine. This was confirmed during the quantification of TSP associated with the indigestible fraction. It was evident that there was a tendency for more TSP to be associated with the insoluble indigestible fraction (IIF) in all treatments, with T3 and T4 having the highest concentration. The same order was observed in the soluble indigestible fraction (SIF), although the TSP concentration associated with this fraction was much lower. IIF and SIF are fractions that remain undigested and unabsorbed after gastrointestinal digestion; this includes dietary fiber, pectin, cellulose, resistant starch, and associated polyphenols [62].

In the study by González-Moya et al. [36], higher TSP content was observed in the IIF, particularly in the treatment containing mango seed extract. As previously mentioned, the association of phenolic compounds with indigestible components may depend on the amount of indigestible fraction in the food matrix, as well as any modifications it has undergone during processing. In this context, yam bean is a food matrix that natively contains a high dietary fiber content, primarily insoluble fiber, which can influence the association of phenolic compounds [63].

It is likely that by the end of the gastrointestinal digestion process, treatments T1 and T4 exhibited low TSP content associated with both the SIF and IIF. Some authors report that the main group of compounds found in tubers are isoflavones, most of which are glycosylated and acetylated via β-glycosidic bonds. The hydrolysis of aglycones occurs throughout the intestinal tract, releasing daidzein, genistein, and glycitein [28]. It has been reported that certain lactic acid bacteria can degrade these compounds through enzymatic action (β-glucosidases) and biotransform them into structures similar to estrogen, mimicking the function of estradiol in the body [64]. However, the amount of TSP associated with these fractions may also be due to vacuum impregnation, which can adhere phenolic compounds to the cell wall of the food due to pressure gradients and the expansion of intercellular spaces [65].

The results of the bioaccessibility percentage (BA) are presented in Table 4, which were statistically different (*p* < 0.05) across all treatments, ranging from 11.47% to 30.65% for T4 and T2, respectively. These values were lower than those reported by González-Moya et al. [36], who reported BA values of 51% to 53%. However, it is important to note that the in vitro digestion conditions used in this study were different. The low BA observed in this study may be related to the binding of phenolic compounds to macromolecules, which hinders their release and absorption (e.g., carbohydrates) [66]. Nevertheless, a low BA could imply that the compounds associated with the indigestible fraction may be metabolized in the colon by colonic microbiota; in this way, the compounds can exert their biological effects upon bioconversion [67].

Conversely, the content of non-bioavailable polyphenols (NBP) was highest in T4 (19.29 mg GAE/g DW), followed by T2, T1, and T3. In the study by González-Moya et al. [36], it was observed that the VI treatment with mango seed extract increased the NBP content compared to the controls. The same effect was observed in this study for T4 and T2. In the case of T1 and T2, the results were similar to those reported by these authors (12.04 and 13.02 mg GAE/g DW). This value is likely to be influenced by the BA of the phenolic compounds; as BA decreases, it is highly probable that the NBP content will increase.

Finally, in the content of potentially bioavailable polyphenols for absorption (PBPA), a generally low concentration was observed, which aligns with the low BA values observed in the treatments. This was particularly evident in T4, where PBPA values were not detected. Interestingly, T2 showed higher values of BA and PBPA compared to T4. This difference could be due to the dilution used in T4, which resulted in a final polyphenol concentration approximately half that of T2. However, another explanation may be the presence of *P. acidilactici* in T4, leading to interactions between the bacteria and phenolic compounds. Such hydrophobic interactions could result in stronger binding of polyphenols to bacterial cell wall components (proteins and lipids), potentially reducing their release and bioavailability in the intestinal phase [68]. This hypothesis is supported by the increase in NBP, where phenolic compounds were higher in T4. The result indicated that, indeed, the treatments subjected to vacuum impregnation exhibited a low absorption capacity, which may be related to vacuum impregnation’s ability to protect them from gastrointestinal digestion. However, the addition of *P. acidilactici* may not be advisable if the primary goal is to maximize the BA and absorption of phenolic compounds, as its presence could hinder the release of polyphenols in the intestinal phase. Nevertheless, the potential probiotic benefits of *P. acidilactici* should also be considered, as they may provide additional health benefits that extend beyond polyphenol absorption.

#### 3.4.2. Changes in Antioxidant Capacity (AOX)

Figure 3 shows the changes associated with the antioxidant capacity (AOX) of the compounds released at each stage of digestion. Overall, a correlation pattern was observed in the three AOX assays with the TSP content (Figure 2). Starting with the ABTS assay (Figure 3a), a marked trend of decreasing AOX was observed from the oral to the intestinal phase, with values in T1 and T3 being very close to zero. The addition of mango seed extract had an influence on T2 and T4, as their AOX in the intestinal stage was higher than the rest of the treatments. In the case of the indigestible fractions, it was again observed that the insoluble indigestible fraction (IIF) had a higher AOX, which even increased in T1, T3, and T4 compared to the undigested food. This might indicate that the AOX measured by the ABTS method is greater in the compounds associated with the indigestible fraction. Additionally, the undigested food showed higher activity by this method, which is consistent with the results observed during in vitro digestion.

In the DPPH assay (Figure 3b), fluctuations were observed across all treatments throughout the digestion process; however, in the intestinal fraction, the values were again close to zero, with only T2 standing out, although the concentration was considered low. Similar to the ABTS assay, the indigestible fractions showed the same AOX distribution pattern, with the highest DPPH values primarily in T2, followed by T3, T1, and T4, all in the IIF.

Finally, in the FRAP assay (Figure 3c), two trends were observed at the beginning of the digestion process after the general decrease that occurs during the oral phase. In the case of T2 and T4, as the process reached the intestinal stage, the phenolic compounds released increased the AOX; conversely, T1 and T3 exhibited an opposite behavior. These trends may indicate that the compounds associated with the addition of mango seed extract had a greater affinity for the mechanisms presented in the FRAP assay. In the indigestible fractions, the IIF showed higher activity in this assay; however, when compared to ABTS and DPPH, it was relatively lower in all treatments.

In the study conducted by González-Moya et al. [36], the highest AOX was also reported in the ABTS assay; however, these were presented in the intestinal fraction. An increase in antioxidant capacity after in vitro gastrointestinal digestion could be due to the expansion of the polysaccharide structures in yam beans when subjected to vacuum impregnation, such that their hydroxyl groups became exposed and able to react and exert antioxidant capacity, in addition to the possible complexes formed with the impregnated phenolic compounds [23]. Furthermore, the increased AOX observed in T3 in the DPPH assay in the IIF could be attributed to the production of exopolysaccharides (EPS) by *P. acidilactici* during gastrointestinal digestion. Previous studies have shown that probiotics, including *Lactobacillus paracasei* subsp. *paracasei* and *Lactobacillus plantarum*, produce EPS with significant antioxidant properties, such as DPPH free radical scavenging, ferrous ion chelation, and inhibition of lipid peroxidation [69]. Further analysis of EPS production by *P. acidilactici* in this system would provide clearer insights into its contribution to the observed antioxidant activity.

#### 3.4.3. Survival of *P. acidilactici*

Figure 4 shows the survival values of *P. acidilactici* during the gastrointestinal digestion process. While T1 and T2 are indicated in the figure, they did not contain any bacterial concentration, and therefore, no CFU was observed throughout the process. For T3 and T4, both treatments began the oral phase without significant differences (*p* < 0.05) and with a concentration close to 10^6^ log of CFU/g. It has been reported that during probiotic intake, these bacteria bypass the mouth and the action of salivary alpha-amylase. However, the action of this enzyme can indirectly benefit probiotics by hydrolyzing carbohydrates that can serve as an energy source and maintain their stability [25]. Once the gastric phase was reached, both treatments experienced a decrease in concentration; T4 showed a decrease of up to three logarithms (10^3^ log of CFU/g), followed by T3 (10^4^ log of CFU/g). When probiotic bacteria enter the stomach, they encounter an extremely acidic environment due to hydrochloric acid. This can be detrimental to some strains, as they may be destroyed or inactivated by the stomach’s acidity. In addition, the enzymatic action of pepsin can break the peptide bonds of the amino acids that make up the bacterial cell wall, permeating the cell membrane [70]. The decrease in the log of CFU/g of food in the gastric phase was also observed in the study by Burca-Busaga et al. [25] in *Lactobacillus salivarius*.

In the intestinal fraction, *P. acidilactici* content stabilized at 10^4^ log of CFU/g in both treatments; however, T3 showed a slightly higher concentration (*p* < 0.05). In this stage, the physiological conditions shift to a less acidic environment, promoting bacterial stability. Additionally, the compounds that were hydrolyzed and released can act as carbon sources to improve the metabolism, viability, and multiplication of the bacteria, as observed in T4 with an increase of one logarithm in bacterial concentration [25]. Regarding the total indigestible fraction (TIF), the concentration remained stable at 10^4^ log of CFU/g in both treatments, with no statistical differences (*p* > 0.05). Within this fraction, probiotic bacteria can access a wide range of carbon sources, which they can utilize when the fermentation process begins. It is important to note that the final concentration for both treatments is above 10^4^ log of CFU/g of food, being close to the minimum concentration of potential probiotic effect [5,71]

## 4. Conclusions

This study evaluated the viability of *Pediococcus acidilactici* and the bioaccessibility of mango seed polyphenols in vacuum-impregnated yam bean snacks following in vitro gastrointestinal digestion. The findings indicate that while vacuum impregnation technology is effective in incorporating functional components such as probiotics and polyphenols into plant-based matrices, the stability and bioaccessibility of these components may be compromised during the digestive process.

The limited antimicrobial activity observed for mango seed extract and *P. acidilactici* may be attributed to the intrinsic resistance of the selected pathogenic bacteria, the low concentration of bioactive compounds in the treatments, and the selected antimicrobial assay. This suggests the potential benefit of increasing the concentration of mango seed extract, optimizing the *P. acidilactici* suspension to enhance efficacy, and correct selection of antimicrobial assay.

Conversely, the mango seed extract demonstrated significant antioxidant capacity once impregnated into yam bean slices, highlighting the effectiveness of vacuum impregnation in enriching food matrices; the relatively low bioaccessibility of the polyphenols suggests increased interaction with indigestible components, leading to reduced release during digestion. *P. acidilactici* exhibited good viability during the impregnation and dehydration processes, though its survival decreased during gastric digestion. Despite this reduction, the final concentration remained close to the threshold considered effective as a potential probiotic dose. These results suggest that future studies could explore the encapsulation of *P. acidilactici* as a strategy to improve its stability and efficacy, ensuring that higher concentrations reach the colon for optimal probiotic benefits. Moreover, the chemical composition of the mango seed extract and the metabolites resulting from the activity of *P. acidilactici* after digestion should be evaluated.

## Figures and Tables

**Figure 1 microorganisms-12-01993-f001:**
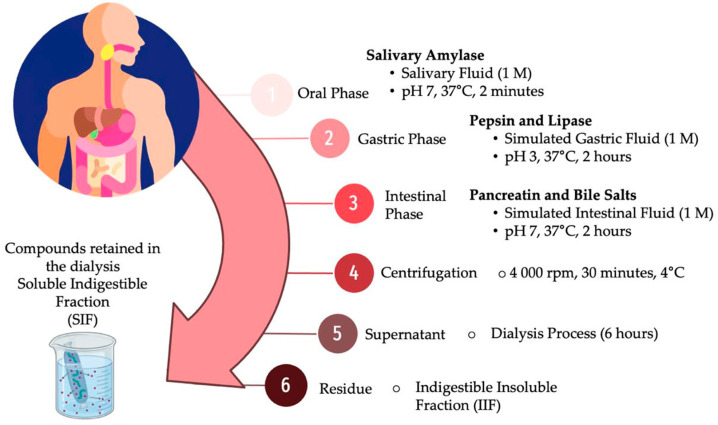
In vitro model of gastrointestinal digestion.

**Figure 2 microorganisms-12-01993-f002:**
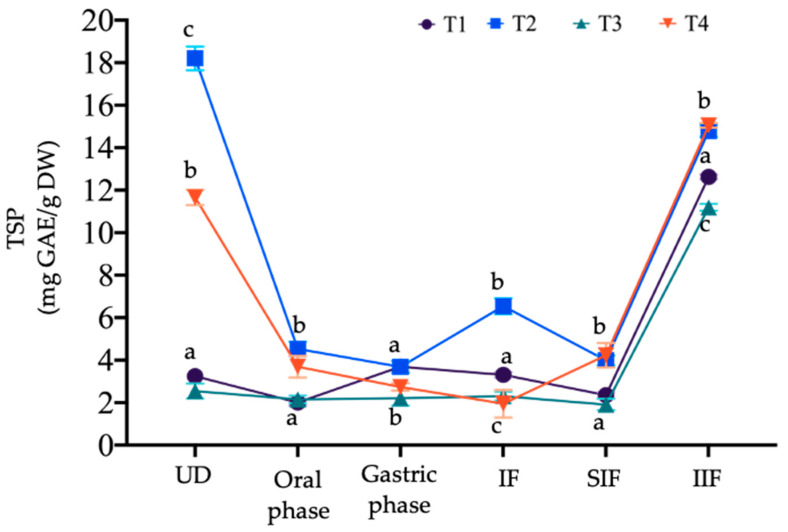
Changes in total soluble phenolic content during in vitro gastrointestinal digestion. Different lowercase letters in each phase/fraction indicate significant differences between treatments (*p* < 0.05). T1—control; T2—mango seed extract at 7 mg/mL; T3—10^6^ cell/mL of *P. acidilactici*; T4—combination, 50:50 *v*/*v* of 7 mg/mL mango seed extract and 10^6^ cell/mL of *P. acidilactici*. UD, undigested; IF, intestinal fraction; SIF, soluble indigestible fraction; IIF, insoluble indigestible fraction.

**Figure 3 microorganisms-12-01993-f003:**
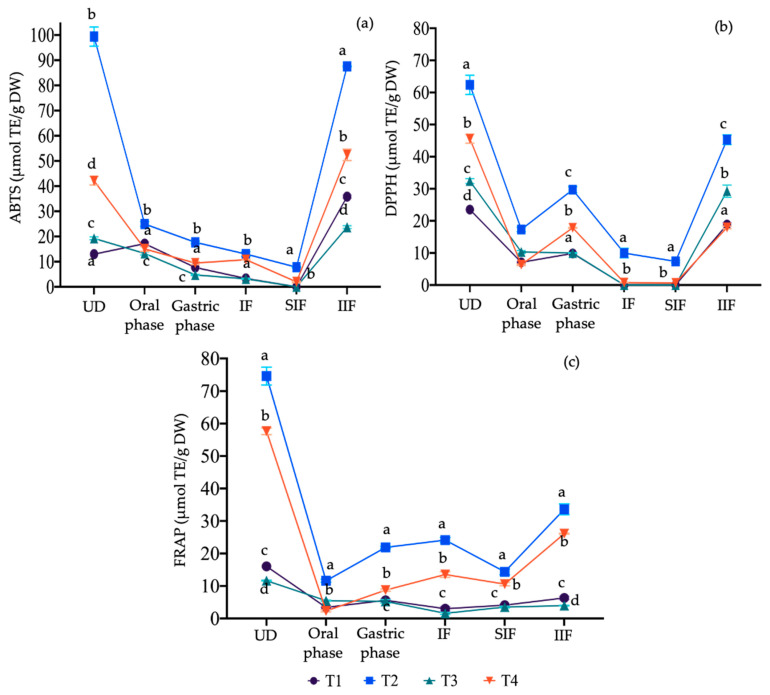
Changes in antioxidant capacity by (**a**) ABTS, (**b**) DPPH, and (**c**) FRAP methods during in vitro gastrointestinal digestion. Different lowercase letters in each column indicate significant differences between treatments (*p* < 0.05). T1—control; T2—mango seed extract at 7 mg/mL; T3—10^6^ cell/mL of *P. acidilactici*; T4—combination (50:50 *v*/*v*) of mango seed extract at 7 mg/mL and 10^6^ cell/mL of *P. acidilactici*. UD, undigested; IF, intestinal fraction; SIF, soluble indigestible fraction; IIF, insoluble indigestible fraction.

**Figure 4 microorganisms-12-01993-f004:**
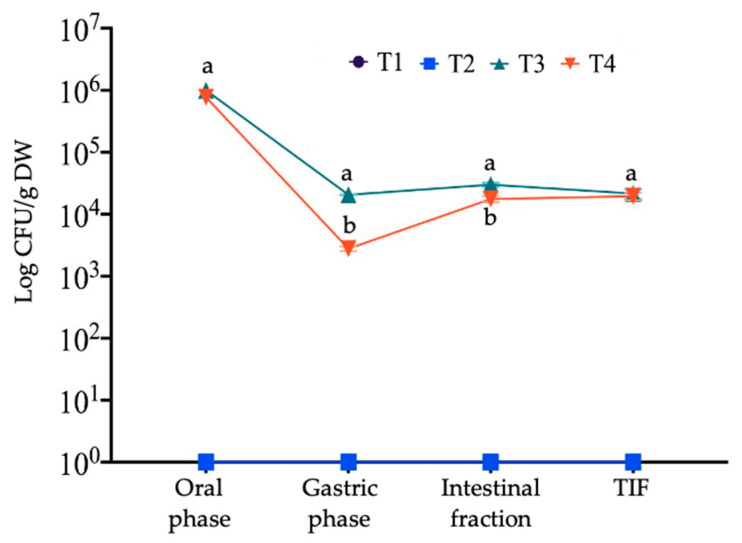
Survival of Pediococcus acidilactici during in vitro gastrointestinal digestion. Different lowercase letters in each column indicate significant differences between treatments (*p* < 0.05). T1—control; T2—mango seed extract at 7 mg/mL mango seed extract; T3—10^6^ cell/mL of *P. acidilactici*; T4—combination (50:50 *v*/*v*) of 7 mg/mL mango seed extract and 10^6^ cell/mL *P. acidilactici*; detection limit (10 CFU/mL of sample).

**Table 1 microorganisms-12-01993-t001:** Sensitivity of pathogenic bacteria to mango seed extract and *P*. *acidilactici* suspension ^1^.

Treatment	Pathogenic Bacteria (Zone Diameter Breakpoint, mm)							
	*S. aureus*		*E. coli*		*P. aeruginosa*		*S. marcescens*	
Mango seed extract (7 mg/mL)	9	R	9	R	10	R	9	R
*P. acidilactici* (106 cell/mL)	0	R	0	R	0	R	0	R
Combination (1:1, *v/v*)	6	R	7	R	7	R	6	R
Gentamicin (10 mg) ^a^	15	S	-		19	S	-	
Tetracycline (30 mg) ^b^	-		20	S	-		-	
Cefotaxime (30 mg) ^c^	-		-		-		20	I

^1^ The diameter breakpoint values are represented in millimeters (mm). The criteria for resistance (R), intermediate resistance (I), and susceptibility (S) were based on the guidelines provided by the Clinical and Laboratory Standards Institute (CLSI) for each control/treatment applied. Treatments were compared with the respective control for each pathogen, and their category was assigned accordingly. The interpretation inhibition zone values (mm) were ^a^ ≥15 (S), 13–14 (I), ≤12 (R); ^b^ ≥15 (S), 12–14 (I), ≤11 (R); ^c^ ≥23 (S), 20–22 (I), ≤19 (R).

**Table 2 microorganisms-12-01993-t002:** Total soluble phenols content, hydrolyzable polyphenols, and antioxidant capacity (ABTS, DPPH, and FRAP) in vacuum-impregnated snacks ^1^.

Treatment	TSP	HP	ABTS	DPPH	FRAP
T1	3.25 ± 0.15 ^a^	4.61 ± 0.14 ^b^	12.59 ± 0.58 ^a^	23.53 ± 1.12 ^a^	16.06 ± 0.64 ^a^
T2	18.21 ± 0.56 ^c^	6.35 ± 0.01 ^a^	99.38 ± 3.82 ^b^	62.37 ± 2.98 ^b^	74.62 ± 2.74 ^b^
T3	2.55 ± 0.35 ^a^	5.17 ± 0.11 ^c^	19.25 ± 0.55 ^c^	32.41 ± 0.76 ^c^	11.65 ± 0.16 ^c^
T4	11.66 ± 0.36 ^b^	6.41 ± 0.12 ^a^	42.14 ± 1.67 ^d^	45.59 ± 1.38 ^d^	57.65 ± 1.03 ^d^

^1^ Values represent mean ± standard deviation (*n* = 3). Different lowercase letters in each column indicate significant differences between treatments (*p* < 0.05). The values for total soluble phenols (TSP) and hydrolyzable polyphenols (HP) are expressed in mg gallic acid equivalents (GAE) per gram of dry weight (DW). ABTS, DPPH, and FRAP values are expressed in μmol Trolox equivalents (TE) per gram of dry weight (DW). T1—control; T2—mango seed extract at 7 mg/mL; T3—10^6^ cell/mL of *P. acidilactici*; T4—combination, 50:50 *v*/*v* of 7 mg/mL mango seed extract and 10^6^ cell/mL of *P. acidilactici*.

**Table 3 microorganisms-12-01993-t003:** Survival of *Pediococcus acidilactici* after vacuum impregnation and dehydration processes ^1^.

Treatment	Vacuum Impregnation	Dehydrated
T1	0	0
T2	0	0
T3	1.60 × 10^6^ ± 2.80 × 10^2 a^	3.70 × 10^6^ ± 2.10 × 10^3 a^
T4	1.90 × 10^6^ ± 1.70 × 10^2 b^	1.75 × 10^6^ ± 4.29 × 10^2 b^

^1^ Values represent mean ± standard deviation (*n* = 3). Different lowercase letters in each column indicate significant differences between treatments (*p* < 0.05). The count of *Pediococcus acidilactici* is represented as colony-forming units (CFU) per gram of dry weight (DW). T1—control; T2—mango seed extract at 7 mg/mL; T3—10^6^ cell/mL of *P. acidilactici*); T4—combination, 50:50 *v*/*v* of 7 mg/mL mango seed extract and 10^6^ cell/mL of *P. acidilactici*; detection limit—10 CFU/mL of sample.

**Table 4 microorganisms-12-01993-t004:** Bioaccessibility percentage (%BA), non-bioavailable polyphenols (NBP), and potentially bioavailable polyphenols for absorption (PBPA) in vacuum-impregnated snacks after in vitro gastrointestinal digestion ^1^.

Treatment	BA (%)	NBP (mg GAE/g DW)	PBPA (mg GAE/g DW)
T1	20.77 ± 0.65 ^a^	14.96 ± 0.15 ^a^	0.96 ± 0.03 ^a^
T2	30.65 ± 1.45 ^b^	18.78 ± 0.06 ^b^	2.55 ± 0.02 ^b^
T3	17.13 ± 0.46 ^c^	13.11 ± 0.42 ^c^	0.39 ± 0.01 ^c^
T4	11.41 ± 1.05 ^d^	19.29 ± 0.55 ^d^	n.d.

^1^ Values represent mean ± standard deviation (*n* = 3). Different lowercase letters in each column indicate significant differences between treatments (*p* < 0.05). Bioaccessibility (BA) is represented as a percentage (%), non-bioavailable polyphenols (NBP), and potentially bioavailable polyphenols for absorption (PBPA) as mg gallic acid equivalents (GAE) per gram of dry weight (DW). T1—control; T2—mango seed extract at 7 mg/mL; T3—10^6^ cell/mL of *P. acidilactici*; T4—combination, 50:50 *v*/*v* of 7 mg/mL mango seed extract and 10^6^ cell/mL of *P. acidilactici*.

## Data Availability

The original contributions presented in the study are included in the article, further inquiries can be directed to the corresponding author.

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
