# Peer review of "In Vitro Digestion of Vacuum-Impregnated Yam Bean Snacks: Pediococcus acidilactici Viability and Mango Seed Polyphenol Bioaccessibility"

_microorganisms, 2024, doi:10.3390/microorganisms12101993_

Round 1

Reviewer 1 Report

Comments and Suggestions for Authors

Reviewer comments: 24.07.2024

A manuscript with the title: “In vitro digestion of vacuum-impregnated yam bean snacks: Pediococcus acidilactici viability and mango seed polyphenol bioaccessibilityby Durán-Castañeda et al., presents in vitro study digestion of vacuum-impregnated yam bean (Pachyrhizus erosus L.), snacks enriched with Pediococcus acidilactici and mango seed polyphenols, aiming to evaluate the survival of the probiotic and the bioaccessibility of the polyphenols.

1. The manuscript is written correctly, showing care in the choice of appropriate style and scientific language.

2. The methodology chosen for the study is not sufficient, and would need to be improved.

3. It would also be useful to add to the literature and improve the discussion as it is not very comprehensive and treats the research problems contained in it superficially.

Main objections

1.     The quantification of total soluble polyphenols by the Folin-Ciocalteu method, is a fairly rapid and efficient method, but has some limitations as it reacts with reducing compounds. At the same time, the content is converted to a single compound, gallic acid. It would be better to isolate the phenolic fraction and use it as an external standard to calculate the total polyphenol content. It would have been much better to use liquid chromatography coupled to mass spectrometry and UV-VIS detectors for quantitative and qualitative determination, something that was lacking in this study.

2.     I will ask the authors to explain why hydrolyzed polyphenols were used in the research, and what sense does it make? Wouldn't it be worth doing it to hydrolyze polyphenols directly from mango seeds? Why are the values ​​so different compared to non-hydrolyzed ones?

3.     Why is the total polyphenol content so different in individual digestion phases? Shouldn't it decrease with subsequent stages of digestion like T1 I T3 without the addition of seed mango polyphenols? Line 13. Yam bean should have a Latin name because it is the first use in the text.

Line 110. “… in a 35:1 weight-to-volume ratio.” The ratio is not clear to me please state how many grams of plant material were used for extraction and in what volume of solvent.

Line 112. “…maintaining the temperature at 60 °C ± 2 °C…” Why was such a high temperature used to extract the polyphenols, were the authors not concerned that they might lose chemicals that are sensitive to temperature?

Line 113. “maintaining…” should be written in capital letters, because it is the beginning of a sentence.

Lines 114 – 116. “This mixture was subjected to sonication using a Hielscher UP400S (Ger-114 many) ultrasonic processor, while maintaining the temperature at 60 °C ± 2 °C with a 115 thermal bath (TERLAB TE-840D) for 8 minutes.” This sentence is a repetition of the previous ones and should be deleted.

Line 119. “…hour at 60 °C…” There is the same problem as with extraction. Such a high temperature can break down temperature sensitive compounds.

Lines 132 – 134. I understand that the authors probably did not have access to the direct producer of this plant, but for better reliability of the scientific study, a detailed geographical location and the specific place from which this plant came should have been given. If there is such a possibility, please provide it.

Lines 187 – 188.  Treatments (250 mg) were mixed…” This sentence is not understandable. Could the authors explain what was taken for the extraction of the polyphenols or the plant material or perhaps the extract obtained using the procedure in section 2.1.

Lines 308 – 309. “Cefotaxime, although tested only against S. marcescens, exhibited an inhibition zone of 20 mm, classified as intermediate resistance.” Please explain why cefotaxime, despite having a 20 mm inhibition zone, was classified as intermediate resistance, while for other compounds, a 15 mm inhibition was sufficient.

Reviewer 2 Report

Comments and Suggestions for Authors

Manuscript 3194489

Journal Microorganisms

Title In vitro digestion of vacuum-impregnated yam bean snacks: Pediococcus acidilactici viability and mango seed polyphenol bioaccessibility

The manuscript entitled “In vitro digestion of vacuum-impregnated yam bean snacks: Pediococcus acidilactici viability and mango seed polyphenol bioaccessibility” describes the design of vacuum-impregnated yam bean snacks enriched with Pediococcus acidilactici and mango seed polyphenols, and the survival of the probiotic bacteria and the bioaccessibility of the polyphenols after a simulated gastrointestinal digestion process. The manuscript is interesting but several parts need substantial improvement. Please follow the comments in the file.

Comments on the Quality of English Language

Moderate changes are necessary

Round 2

Reviewer 1 Report

Comments and Suggestions for Authors

Thank you, for responding to my review and addressing the suggestions in it. This revised version can be accepted for publication

Reviewer 2 Report

Comments and Suggestions for Authors

Authors addressed the reviewer's comments in the revised manuscript. Minor comments are reported below:

Fig. 3  Why in the DPPH assay T3 sample showed a different behaviour in comparison to ABTS and FRAP assays (see IIF) and the other samples? Please discuss this result in the section 3.4.2

Fig. 4 Add the limit of detection in the caption

Comments on the Quality of English Language

Minor revisions are necessary
